# Quark formation and phenomenology in binary neutron-star mergers using V-QCD

Samuel Tootle,[1] Christian Ecker,[1] Konrad Topolski,[1] Tuna Demircik,[2] Matti Järvinen,[2, 3] and Luciano Rezzolla[1, 4, 5]

[1]*Institut für Theoretische Physik, Goethe Universität, Max-von-Laue-Str. 1, 60438 Frankfurt am Main, Germany*
[2]*Asia Pacific Center for Theoretical Physics, Pohang, 37673, Korea*
[3]*Department of Physics, Pohang University of Science and Technology, Pohang, 37673, Korea*
[4]*Frankfurt Institute for Advanced Studies, Ruth-Moufang-Str. 1, 60438 Frankfurt, Germany*
[5]*School of Mathematics, Trinity College, Dublin 2, Ireland*
(Dated: May 18, 2022)

Using full 3+1 dimensional general-relativistic hydrodynamic simulations of equal- and unequal-mass neutron-star binaries with properties that are consistent with those inferred from the inspiral of GW170817, we perform a detailed study of the quark-formation processes that could take place after merger. We use three equations of state consistent with current pulsar observations derived from a novel finite-temperature framework based on V-QCD, a non-perturbative gauge/gravity model for Quantum Chromodynamics. In this way, we identify three different post-merger stages at which mixed baryonic and quark matter, as well as pure quark matter, are generated. A phase transition triggered collapse already $\lesssim 10\,\mathrm{ms}$ after the merger reveals that the softest version of our equations of state is actually inconsistent with the expected second-long post-merger lifetime of GW170817. Our results underline the impact that multi-messenger observations of binary neutron-star mergers can have in constraining the equation of state of nuclear matter, especially in its most extreme regimes.

Keywords: neutron star mergers, critical point, gauge/gravity duality

## I. INTRODUCTION

Multi-messenger observations from binary neutron-star mergers are expected to provide new insights into the properties of dense and hot Quantum Chromodynamics (QCD). However, due to the lack of first principle techniques, the precise phase structure of QCD and its equation of state (EOS) are currently not known at densities and temperatures that occur during and after the merger of two neutron stars. It is, therefore, not clear if a transition from dense nuclear to quark matter can happen in such events nor if such a transition leaves a discernible imprint on the gravitational waveform or the electromagnetic counterpart. Such an imprint may be best visible in the kHz gravitational-wave signal emitted during a possible post-merger hypermassive neutron star (HMNS) stage as it encodes information on the hot and dense part of the EOS that is inaccessible during the inspiral phase, however, this parameter space will only be able to be reliably explored by future detectors. In order to resolve the question of the detectability of quark matter in binary neutron-star mergers, it is therefore crucial to explore state-of-the-art models that make predictions for the phase structure of QCD beyond the saturation density of atomic nuclei $n_s = 0.16\,\mathrm{fm}^{-3}$ and, at the same time, satisfy known constraints from theory and astrophysical observations.

Numerical simulations of binary neutron-star systems play a critical role to ascertain the influence of the neutron-star characteristics (mass and spin) and the EOS on the post-merger remnant and lifetime [1, 2]. For EOSs that include a transition to quark matter, these simulations also provide insights into the mechanisms for and the abundance of quark formation in the post-merger remnant. There exists a number of works that study mergers using models with a phase transition either in an effective particle approach [3] or in fully general-relativistic hydrodynamics [4–6]. The recent study [7] also analyses the composition of the merger ejecta.

In [8] possible signatures of phase transitions in the gravitational wave signal have been classified using a polytropic EOS with the phase transition to quark matter introduced via a Gibbs-like construction and temperature dependence using the standard $\Gamma_{\mathrm{th}}$ component based on an ideal-fluid EOS [9].

In this article we perform the first study to analyse quark matter production in BNS mergers using a novel framework [10] for describing nuclear matter which combines the gauge/gravity duality with nuclear theory models. In this combined framework, the EOS at large densities is given by the gauge/gravity duality [11, 12], more precisely the V-QCD model [13–15], which is used to describe the deconfinement phase transition from dense baryonic to quark matter. The DD2 version of the Hempel-Schaffner-Bielich statistical model [16, 17] is used at low densities, and a van der Waals model for the temperature dependence in the dense nuclear matter phase. The advantage of this combined framework is that it is, by construction, consistent with theoretic predictions from nuclear theory at low densities and perturbative QCD at high densities. At the same time, it provides a description for the deconfinement phase transition within a single model, i.e., V-QCD. The framework also yields predictions for the location of the QCD critical point which allows us to investigate the possibility that neutron star matter during and/or after the merger reach temperatures sufficient to probe this part of the QCD phase diagram.

When taking into account different individual masses and spins of the stars as well as variations of the EOS, the parameter space of binary neutron star simulations grows rapidly. As a first test of the novel V-QCD EOS, we study configurations motivated by information gained from the inspiral part of the gravitational-wave event GW170817 [28] in order to determine whether or not the constructed EOSs agree with the observation. To do so, we perform full 3+1 dimensional general-relativistic hydrodynamics (GRHD) simulations consistent with GW170817 so as to determine mechanisms for quark production as well as ascertain the importance of multi-

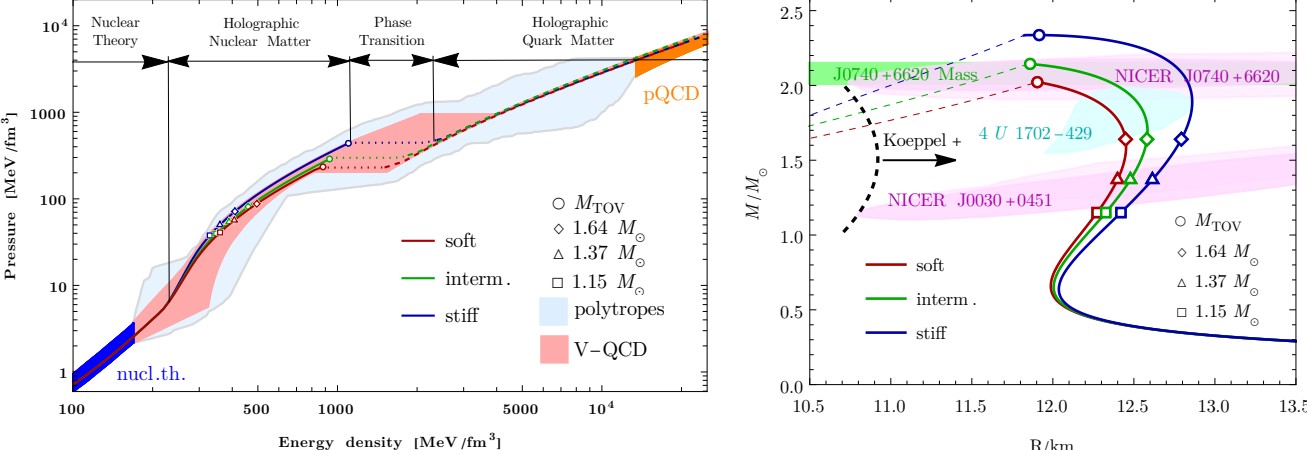

FIG. 1. Left: Cold, beta equilibrium slices of the V-QCD EOS. Shown are the `Soft` (red), `Inter` (green) and `Stiff` (blue) versions of the model. Dark blue and orange bands mark the uncertainties in nuclear theory [18] and perturbative QCD [19] calculations, respectively. The light blue region marks the theoretically allowed region spanned by polytropic EOSs, where the pink region marks the parametric freedom of V-QCD after imposing constraints from NS observations. Right: Corresponding mass-radius relations where dashed lines mark unstable quark matter branches. In addition we show various error bands of direct mass [20, 21] and radius [22–26] measurements of heavy pulsars; shown with a black dashed line is a theoretical lower bound for the radii as computed from considerations on the threshold mass by Koeppel et al. [27].

messenger detections on constraining the model. More specifically, we simulate non-spinning binaries of fixed chirp mass $\mathcal{M}_{\mathrm{chirp}} := (M_1 M_2)^{3/5}/(M_1 + M_2)^{1/5} = 1.186\,M_\odot$ and two different mass ratios $q := M_2/M_1 = 1, 0.7$. We use three different EOSs, based on a soft, an intermediate and a stiff version of V-QCD, whose cold beta-equilibrium parts satisfy [29–31] all currently known observational constraints from direct mass [20, 21, 32] and radius [22, 23, 25, 26] measurements of heavy pulsars as well as the constraint on the tidal deformability obtained from GW170817 [33].

As an important outcome of our simulations we identify three different stages of quark matter formation during the post-merger HMNS evolution which will be referenced as hot-quark (HQ), warm-quark (WQ) and cold-quark (CQ) stages. The basis and definition of these stages are discussed in detail in Section IV. Furthermore, we provide an analysis of the gravitational waveforms and their frequency spectra of all our simulations. The clearest imprint on the waveform is their early termination caused by the phase-transition-triggered collapse to a black hole in the CQ stage. From their frequency spectra we extract a number of characteristic post-merger frequencies.

The rest of the article is structured as follows. In Section II we provide more details about the V-QCD model. In Section III we explain our numeric setup for the binary initial data and the subsequent time evolution. In Section IV we present our results for the merger dynamics and show examples for the three post-merger stages mentioned earlier. In Section V we discuss our results for the gravitational waves and their spectra. In Section VI we summarise and conclude. Finally, in Appendix A we show results obtained with lower resolution as in the main text and study the consistency of our simulations. Hereafter, we use units where we set $k_{\mathrm{B}} = G = c = 1$ unless otherwise noted.

## II. MATTER MODEL

In this section we provide a brief summary of the combined framework used in our merger simulations. The most important novel ingredient in this framework is the use of the gauge/gravity duality. The gauge/gravity duality allows one to translate intractable problems in quantum field theory at strong coupling and finite density to tractable problems in classical five-dimensional gravity. This approach has been successfully applied in the context of heavy-ion collisions [34, 35], condensed matter physics [36, 37] and, more recently, also astrophysics [38–43]. The specific gauge/gravity model which we are using is V-QCD, which we will now briefly review, however, a detailed introduction to V-QCD and to the EOS at finite temperature and charge fraction based on V-QCD can be found in [10, 40].

V-QCD includes sectors both for quarks [44, 45] and for gluons [46, 47]. It is based on a string theory setup in the Veneziano limit [48, 49] in which both the number of colors $N_c$ and flavors $N_f$ are large, but their ratio is $\mathcal{O}(1)$ [13, 50, 51] such that the quark degrees of freedom strongly back-react to the gluon dynamics. However, the model cannot be strictly derived from string theory, and one switches to an effective "bottom-up" approach where the action of the model is tuned to agree with QCD data at finite $N_c$ and $N_f$. Therefore the model includes a relatively large number of parameters that are tuned to reproduce known features of QCD such as asymptotic freedom, confinement, linear glueball and meson trajectories and chiral symmetry breaking. The remaining freedom is constrained by lattice data for large-$N_c$ pure Yang-Mills theory [52] and data for $N_c = 3$ QCD with $N_f = 2 + 1$ flavors at physical quark masses [53, 54] at small baryon-number density. Such an effective approach turns out to be useful to

| | $q$ | $M_{\rm TOV}$ $[M_\odot]$ | $M_1$ $[M_\odot]$ | $M_2$ $[M_\odot]$ | $R_1$ [km] | $R_2$ [km] | $\Lambda_1$ | $\Lambda_2$ | $\tilde\Lambda$ | $\Lambda_{1.4}$ | $f_{\rm mer}$ kHz | $f_3$ kHz | $f^{2,1}$ kHz | $f^{2,2}$ kHz | $t_{\rm BH}$ [ms] |
|---|---|---|---|---|---|---|---|---|---|---|---|---|---|---|---|
| `Soft_q10` | 1.0 | 2.02 | 1.37 | 1.37 | 12.37 | 12.37 | 537 | 537 | 537 | 475 | 1.77 | 4.00 | 1.44 | 2.92 | 9.5 |
| `Soft_q07` | 0.7 | 2.02 | 1.64 | 1.15 | 12.42 | 12.24 | 183 | 1393 | 517 | 475 | 1.63 | 3.80 | 1.55 | 2.80 | 5.8 |
| `Soft_q10-NPT` | 1.0 | 2.06 | 1.37 | 1.37 | 12.37 | 12.37 | 537 | 537 | 537 | 475 | 1.76 | 4.00 | 1.62 | 2.85 | > 37 |
| `Soft_q07-NPT` | 1.0 | 2.06 | 1.64 | 1.15 | 12.42 | 12.24 | 183 | 1393 | 517 | 475 | 1.64 | 3.85 | 1.47 | 2.79 | 11 |
| `Inter_q10` | 1.0 | 2.14 | 1.37 | 1.37 | 12.45 | 12.45 | 565 | 565 | 565 | 511 | 1.74 | 4.00 | 1.51 | 2.73 | > 35 |
| `Inter_q07` | 0.7 | 2.14 | 1.64 | 1.15 | 12.56 | 12.30 | 201 | 1437 | 543 | 511 | 1.63 | 3.70 | 1.42 | 2.67 | > 37 |
| `Stiff_q10` | 1.0 | 2.34 | 1.37 | 1.37 | 12.58 | 12.58 | 617 | 617 | 617 | 560 | 1.74 | 3.90 | 1.39 | 2.49 | > 37 |
| `Stiff_q07` | 0.7 | 2.34 | 1.64 | 1.15 | 12.76 | 12.38 | 231 | 1525 | 591 | 560 | 1.59 | 3.50 | 1.39 | 2.48 | > 38 |

TABLE I. Properties of cold non-spinning isolated neutron stars used in the construction of the binary initial data. Specifically, for each model considered, we list the mass ratio, $q$; the maximum non-rotating mass, $M_{\rm TOV}$; the gravitational mass of each object at infinite separation, $M_{<1/2>}$; the proper radius of each object, $R_{<1/2>}$; the tidal deformability of each object, $\Lambda_{<1/2>}$; the binary tidal deformability, $\tilde\Lambda$; the tidal deformability of a $1.4\,M_\odot$ NS, $\Lambda_{1.4}$; the instantaneous frequency at the maximum GW strain amplitude, $f_{\rm mer}$; the characteristic frequencies of the post-merger phase, $f_3$, $f^{2,1}$, $f^{2,2}$; and the collapse time to black hole formation, $t_{\rm BH}$, for the configurations that collapsed within the evolution time.

model dense nuclear and quark matter in the region where the QCD-coupling is large and traditional nuclear theory calculations and perturbation theory are not reliable [54–56].

In the current version of the model baryons are implemented in a homogeneous approximation [14]. This approximation is natural at densities well above $n_s$, where the distance between neighbouring nucleons becomes smaller than their diameter and a description in terms of homogeneous matter is expected to work well. At densities below $n_s$ this homogeneous approximation breaks down and traditional nuclear matter models become more reliable. In the combined framework this is taken into account by constructing hybrid EOSs whose cold low-density part is modelled by nuclear theory and the dense baryonic and quark matter part via the gauge/gravity duality. For the cold nuclear theory part we use a combination [57] of the Hempel-Schaffner-Bielich (HS) EOS [17] with DD2 relativistic mean field theory interactions [16] and the Akmal-Pandharipande-Ravenhall (APR) model [58].

The prediction of V-QCD for the EOS in the nuclear matter phase suffers from a generic limitation of gauge/gravity duality: the temperature dependence is trivial. In [10] this issue was solved by using a simple van der Waals model, i.e., gas of nucleons, electrons and mesons with excluded volume correction for the nucleons and an effective potential. The van der Waals model was chosen to match with the V-QCD prediction for the cold EOS of nuclear matter and then used to extrapolate the result to finite temperature. The model was further improved to allow deviation from beta-equilibrium by imposing the charge fraction dependence of the HS(DD2) model [16, 17] for nuclear matter and a simple model of free electrons for quark matter. In the regime of low density, both the dependence on the temperature and the charge fraction are following the HS(DD2) model. Finally, our framework contains a strong first order phase transition between the nuclear and quark matter phases at low temperatures which ends on a critical point at finite temperature and density. Consequently, there is a mixed nuclear-quark matter phase which was obtained in [10] by carrying out a Gibbs construction depending on all the three variables, i.e., the density, the temperature, and the charge fraction. Surface tension of the nuclear

to quark matter interface is neglected. This is expected to be a good approximation at the temperatures that we encounter in our simulations, because the latent heat at the transition is very high, comparable to the free energy density of the nuclear matter phase.

We here use three variants of the EOS, which were established in [10] in order to represent the parameter dependence of the model. These are called the soft (`Soft`), intermediate (`Inter`), and stiff (`Stiff`) variants; and reflect three different choices of the parameters in the action of V-QCD [54]. In Fig. 1 (left) we show cold beta-equilibrium slices of these three EOSs with uncertainty bands from nuclear theory [18] (blue) and perturbative QCD [19] (orange). Red, green and blue lines are the `Soft`, `Inter` and `Stiff` versions of V-QCD where the dotted part of these curves mark the first order phase transition between the baryonic (solid) the quark matter (dashed) phase. In addition we show markers for the central densities reached in various isolated non-rotating stars that we use to initialise the binary systems in our simulations. Light blue and pink regions mark the residual freedom of polytropic parametrizations of the EOS [59, 60] and of the V-QCD model [29], respectively, after imposing constraints from neutron star observations. In Fig. 1 (right) we show the corresponding mass-radius relations of non-rotating stars together with various error bands of the direct mass measurement (green area) of the heaviest known pulsar PSR J0740+6620 [20, 21] ($M = 2.08 \pm 0.07 M_\odot$) and direct radius measurements of PSR J0740+6620 [25, 26] and PSR J0030+0451 [22, 23] obtained by the NICER experiment (pink ellipses) as well as from the measurement of the X-ray binary 4U 1702-429 [24] (cyan area; see [31] for a more detailed analysis of the impact of the NICER results). Finally, shown with a black dashed line is a theoretical lower bound for the radii as computed from considerations on the threshold mass by Koeppel et al. [27]. We note that we have checked that our EOSs respect the upper bound on the binary tidal deformability $\tilde\Lambda < 720$ (low-spin priors) obtained from GW171817 and are well within the bounds recently proposed on general parameterization of the sound speed in neutron stars [61] (see Table I).

## III.  INITIAL DATA AND EVOLUTION SETUP

To generate the binary neutron-star initial data we use the recently developed Frankfurt University/Kadath (`FUKA`) [62] solver library. `FUKA` is based on an extended version of the `KADATH` spectral solver library [63], which has been specifically designed for numerical relativity applications. To generate the initial data for compact binaries, `FUKA` uses the eXtended Conformal Thin Sandwich formulation of Einstein's field equations [62, 64, 65]. Initially `FUKA` solves the initial data using force-balance equations to obtain a binary in the quasi-circular orbit approximation. In an effort to minimize the residual eccentricity during the inspiral, an eccentricity reduction step is performed using estimates for the orbital- and radial-infall- velocities at 3.5-th post-Newtonian order. As shown in [62], this procedure leads to a significant reduction of the eccentricity in asymmetric and spinning compact object binaries. We find the advanced handling of eccentricities in `FUKA` essential to obtain accurate initial data for the unequal mass binaries studied in this work.

For the binary evolution we make use of the `Einstein Toolkit` [66] infrastructure that includes the fixed-mesh box-in-box refinement framework `Carpet` [67]. For the simulations presented in the main text we use six refinement levels with finest grid-spacing of $\Delta_{\mathrm{H}} := 221 \, \mathrm{m}$. To check the consistency of our results we have performed a series of simulations for the `Soft` EOS with low ($\Delta_{\mathrm{L}} := 369 \, \mathrm{m}$) and medium ($\Delta_{\mathrm{M}} := 295 \, \mathrm{m}$) resolution which is discussed in Appendix A. For the spacetime evolution we use `Antelope` [68], which solves a constraint damping formulation of the Z4 system [69, 70]. Furthermore, to evolve the hydrodynamic part we use the Frankfurt/Illinois (`FIL`) general-relativistic magnetohydrodynamic code [68], which is based on the `IllinoisGRMHD` code [71]. `FIL` implements fourth order conservative finite-differencing methods [72], enabling a precise hydrodynamic evolution even at low resolution. Furthermore, `FIL` is able to handle tabulated EOSs that are dependent on temperature and electron-fraction. In principle, `FIL` also includes a neutrino leakage scheme that can handle neutrino cooling and weak interactions. However, in its current form V-QCD does not include a description for neutrinos which is why we do not make use of the leakage scheme in this work.

Finally, in Table I we summarise the properties of cold non-spinning isolated neutron stars used in the construction of the binary initial data as well as various characteristic frequencies extracted from the post-merger waveforms presented in Sec. V.

## IV.  MERGER DYNAMICS AND QUARK FORMATION

In this section we present the merger dynamics and discuss the three mechanisms in which quarks are formed during the post-merger evolution. We have performed numerical simulations of BNS systems using mass ratios $q = 1, 0.7$ where the neutron stars have an irrotational fluid profile. For each configuration, the initial data and evolution have been computed using the `Soft`, `Inter`, and `Stiff` EOSs in order to compare the influence of stiffness on the quark production and lifetime of the remnant HMNS whose lifetime is expected to be on the order of one second [73, 74]. A comparative overview of the results from these numerical simulations can be seen in Fig. 2 where we show the evolution of the normalized maximum density, $(n_{\mathrm{b}}/n_{\mathrm{s}})_{\mathrm{max}}$, the maximum temperature, $T_{\mathrm{max}}$, and the quark fraction, $\langle Y_{\mathrm{quark}} \rangle$. To further characterise the distribution of quark matter within the remnant HMNS, we define three regions that follow the centre-of-mass of the HMNS in the orbital plane. We will refer to these regions from now on as the inner core ($R < 3 \, \mathrm{km}$), outer core ($3 \, \mathrm{km} < R < 6 \, \mathrm{km}$) and inner crust ($6 \, \mathrm{km} < R < 9 \, \mathrm{km}$). An illustration along with further details regarding the motivation behind these choices is discussed in Appendix C. The quark fraction $Y_{\mathrm{quark}}$ is currently computed in the post-analysis of the evolutions. In order to obtain the associated grid function values we use a multi-linear interpolation scheme for the given tabulated EOS of a respective model, in which discrete data points $Y_{\mathrm{quark}} = Y_{\mathrm{quark}}(n_{\mathrm{b}}, T, Y_{\mathrm{e}})$ are stored, as well as the 2D slices of the $n_{\mathrm{b}}$, $T$ and $Y_{\mathrm{e}}$.

In all three models, comparing $\langle Y_{\mathrm{quark}} \rangle$ to the maximal temperature shows a clear correlation between the appearance of quark matter and the increase in temperature due to shock-heating during the initial merger stage which is coloured red in Fig. 2. For the remainder of this work, we will reference this as hot quark (HQ) production. The amount of HQs produced is significantly larger for the `Soft` EOS than for the stiffer models which is not unexpected as the densities inside a neutron star with fixed mass is larger when matter is soft and is, therefore, easier to produce strong shocks during the collision. Furthermore, the amount of HQs produced is significantly larger in the unequal mass case than in the equal mass case because the initial density of the heavier companion is higher before merger in unequal binaries which subsequently results in a more violent collision and stronger shocks. Another significant difference between equal and unequal mass binaries is that in the latter HQs are formed not only in the inner core (and a tiny amount in the outer core), but also in the outer core and inner crust region. This has to do with the fact that the distribution of matter in the unequal mass case is highly non-axisymmetric which leads to HQs in the hot off-central region of the HMNS.

In the case of the `Soft` EOS, we see during the cooling of the remnant and as the core becomes more dense an additional quark production channel occurs. We will designate this production mechanism as warm quarks (WQ). The formation of WQs appears to be a result of the complicated post-merger dynamics that leads to a periodic expansion and contraction of the HMNS. During the contraction the density increases close to the rotation axis and leads to the formation of quarks. In the subsequent expansion the density decreases and quarks transition back to baryonic matter. In the `Soft` model we find that WQs are produced in the equal mass case $\approx 2 \, \mathrm{ms}$ after the end of the HQ production, while for unequal masses we find a continuous transition between the production of HQs and WQs. For the stiffer models we do not find WQs, however, we expect their formation to be possible with a heavier binary

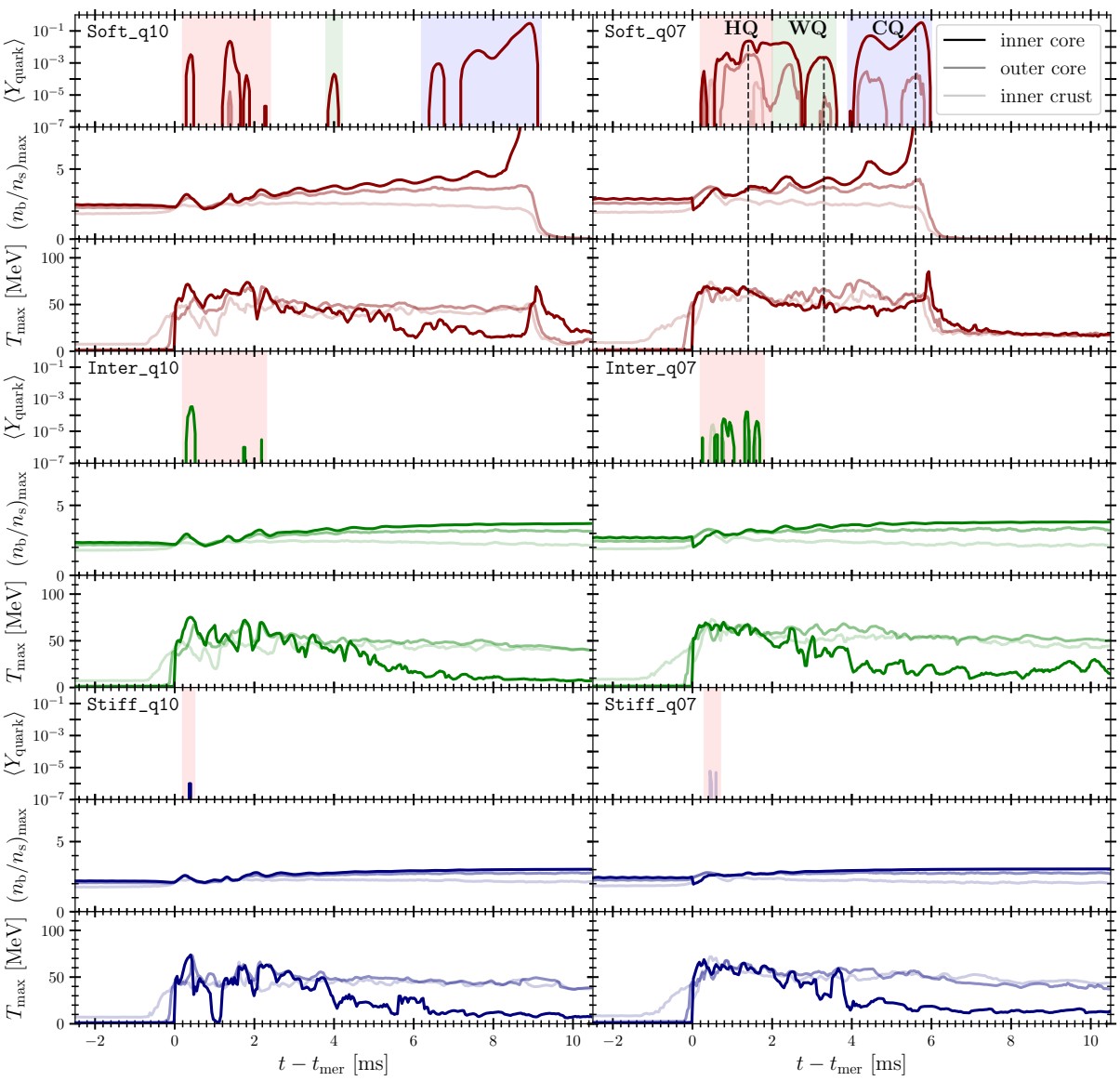

FIG. 2. Correlation between the maximal temperature, maximal density and quark fraction in the three regions of interest, for all three equations of state and both mass ratios. We mark the time slices used for Fig. 3 with vertical black lines. The abbreviations HQ, WQ and CQ stand for the hot-quark, warm-quark and the cold-quark stages respectively.

which will be studied thoroughly in a future work. To capture accurately the delicate combination of temperature and density required for WQs, the use of high resolution numerical methods is essential. This is demonstrated in Appendix A where we compare our results with two lower resolutions for the `Soft` model and find that WQs can be absent in our intermediate and low resolution simulations.

Finally, when the HMNS becomes approximately circular and its compactness increases, a cold but dense quark matter core can be formed which we denote as cold quark (CQ) production. We find CQ production to be present only for the `Soft` model, which ultimately undergoes a phase-transition-triggered collapse for the configurations considered. Notice that due to the large latent heat in the V-QCD model, no sta-

ble quark matter cores can be formed and therefore formation of a quark core inevitably leads to the collapse of the remnant. The starting time of CQ production depends on the mass ratio and the total mass of the system. For the unequal-mass binary it starts already $\approx 4\,\mathrm{ms}$ after merger-time, while in the equal-mass case the starting time is approximately two milliseconds later ($\approx 6\,\mathrm{ms}$). A close inspection of the maximum number density in the core of the unequal-mass binary reveals that CQs set in during the fifth bounce of the merger remnant. In this stage the maximal temperature in the core is lower than in the exterior, in particular for the equal-mass binary. We find this stage to last for $\approx 4\,\mathrm{ms}$ ($\approx 2\,\mathrm{ms}$) for the equal-(unequal-)mass binary, before the density and temperature in the core eventually start to increase rapidly leading to the formation of

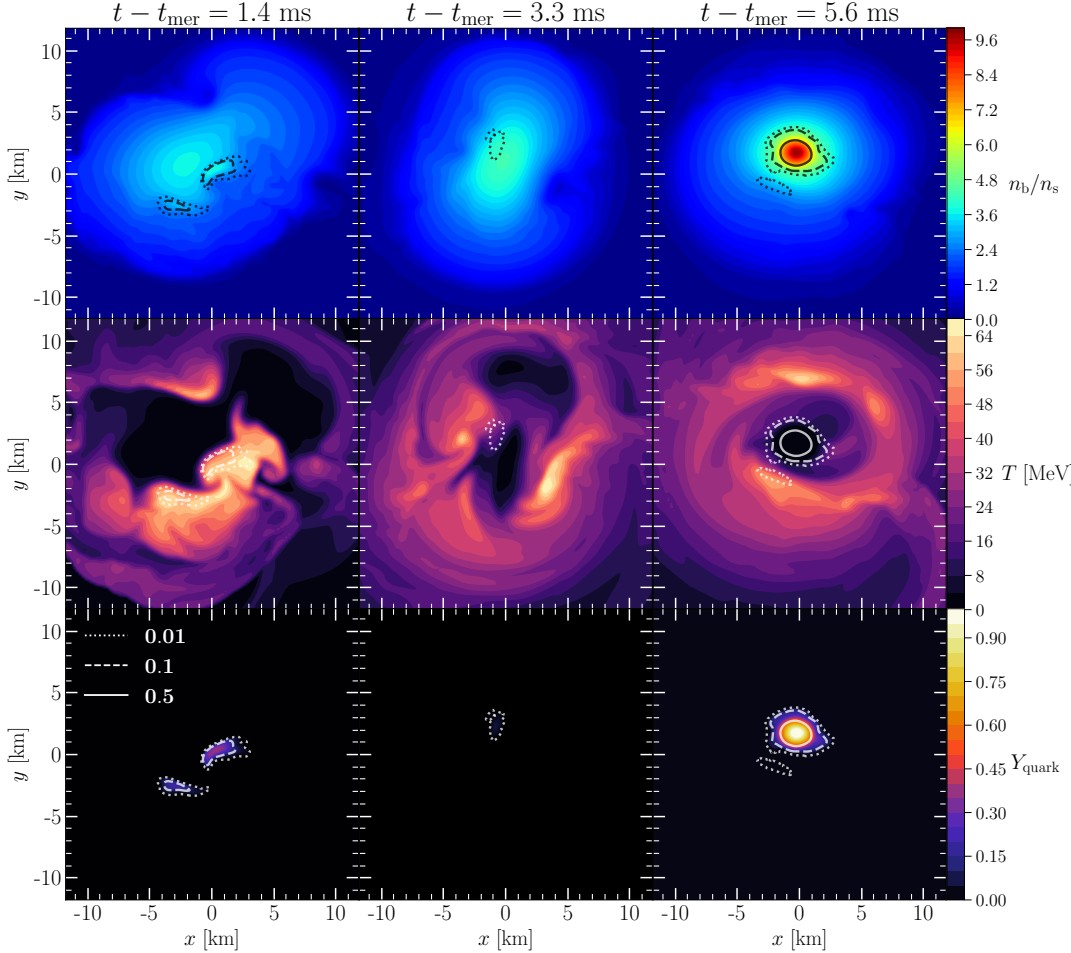

FIG. 3. From top to bottom rows we show snapshots of the baryon-number density, temperature and quark fraction in the orbital plane at three different times $t - t_{\mathrm{mer}} = 1.4, 3.3, 5.6\,\mathrm{ms}$ (from left to right) that are representative for HQ, WQ and CQ stages in the Soft_q07 simulation. In addition we mark contours for the quark fraction $Y_{\mathrm{quark}} = 0.01, 0.1, 0.5$ by dotted, dashed and solid lines, respectively.

a black-hole horizon.

To further explore the appearance of HQ, WQ and CQ we show in Fig. 3 snapshots of the number density (top), temperature (middle) and quark fraction (bottom) in the orbital plane at three representative subsequent times (left to right) of the post-merger evolution of the Soft_q07 configuration. The times have been chosen based on the times of significant HQ, WQ, and CQ production, respectively, as denoted by vertical dotted lines in Fig. 2. Additionally in Fig. 3 we indicate by dotted, dashed and solid lines the outer contours of regions that contain quark fractions larger than 0.01, 0.1 and 0.5, respectively. The plots for $t - t_{\mathrm{mer}} = 1.4\,\mathrm{ms}$ clearly show the presence of HQs in the hottest regions outside the dense core of the HMNS. Typical for this early post merger state is the formation of multiple disconnected hot pockets with temperatures well above $50\,\mathrm{MeV}$ inside the HMNS which, in this case, result in two disconnected regions where HQs are formed. The second column shows snapshots at $t - t_{\mathrm{mer}} = 3.3\,\mathrm{ms}$ corresponding to the formation of a single patch of WQs. We note that a precise definition of the WQ stage is difficult since it sensitively depends on the combination of densities and tem-

peratures outside the hottest and densest regions of the star such that a transition to a mixed-phase is possible. These conditions are typically realized in regions outside, but close to the center of the HMNS where both temperature and density are significantly below their maximal values. Finally, in the third column we show snapshots at $t - t_{\mathrm{mer}} = 5.6\,\mathrm{ms}$ where a pure quark matter core ($Y_{\mathrm{quark}} = 1$) has already been formed and ultimately leads to a phase-transition-triggered collapse of the HMNS. The maximal density of the quark matter core is well above $9\,n_s$ (and rising), but the temperature is very low further motivating our classification of quark matter in this region as CQs. Also noteworthy is the appearance of a small disconnected portion of WQs that forms in a small region outside the dense and cold centre. It is interesting to compare our cold quark core to [75], who find the quark core to be hot. The reason for this difference is that [75] models the temperature dependence by adding a gamma-law to the cold EOS model where the temperature simply scales with the density and does not take into account the change in composition as in V-QCD.

To study the composition of matter in terms of the maxima of $\langle Y_{\mathrm{quark}} \rangle$ compared to $T_{\mathrm{max}}$ and $n_b^{\mathrm{max}}$, we show in Fig. 4

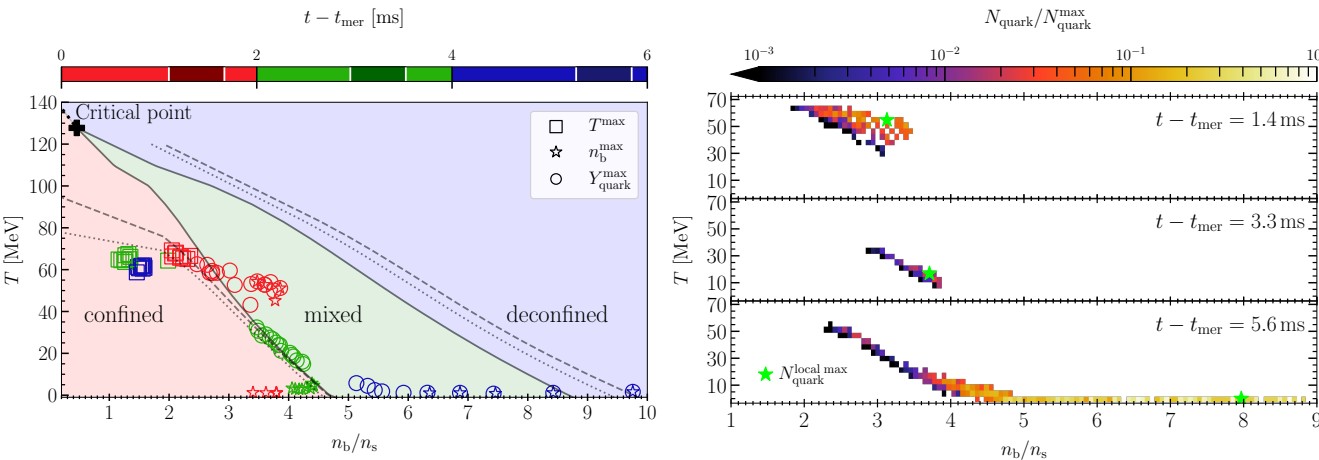

FIG. 4. Left: Phase diagram of the Soft_q07 post-merger evolution, with regions of confined, mixed and deconfined phases of matter shown. Points with a specific colour all occur within time intervals indicated by darker colour bars. Right: The amount of quarks in the orbital plane, measured using the same times as in Fig. 3. We normalize the measured amount of quarks by the maximal amount found in all three time slices presented here.

(left) a phase diagram of the Soft V-QCD model in the temperature and baryon-number density plane where red, green and blue regions indicate confined, mixed and deconfined phases, respectively; and solid lines mark the phase boundaries in beta equilibrium which end at a critical point marked by a black cross. Additionally, we show as a reference dashed and dotted lines indicating the phase boundaries at approximately the minimal ($Y_e = 0.05$) and maximal ($Y_e = 0.09$) values of the charge fraction inside the HMNS during the evolution. These values differ from that of the beta equilibrium, because we determine $Y_e$ by demanding beta equilibrium inside the stars that are used to initialize the binary system, but advect $Y_e$ during the subsequent evolution thereby driving the fluid out of beta equilibrium. Finally, colored symbols mark representative results of the $q = 0.7$ simulation for the maximal temperature (squares), baryon-number density (stars) and quark fraction (circles) collected during the early HQ stage ($t - t_{\mathrm{mer}} = 0 - 2\,\mathrm{ms}$, red), the WQ stage ($t - t_{\mathrm{mer}} = 2 - 4\,\mathrm{ms}$, green) and the CQ stage ($t - t_{\mathrm{mer}} = 4 - 6\,\mathrm{ms}$, blue). From the red symbols one clearly sees the maxima of the quark fraction appearing close to the maxima in temperature at approximately 60 MeV, while the maxima in the number density appear at much lower temperature at the bottom of the phase diagram, illustrating our motivation to classifying these quarks as HQs. The green symbols represent examples for the WQ stage where $Y_{\mathrm{quark}}$ is maximal at intermediate temperature ($\approx 20\,\mathrm{MeV}$) and where the maximal density ($\approx 4\,n_s$) occurs. The maximal temperature is significantly higher ($\approx 60\,\mathrm{MeV}$) in this case, but appears at much lower densities ($\approx 1\,n_s$). Furthermore, the blue symbols mark the CQ production where the maximal quark fraction coincides with the maximum number density at very low temperature.

Finally, in Fig. 4 (on the right) we show our measurements of the amount of quark matter during the three separate time-slices used in Fig. 3. For this we define $126 \times 35$ bins with equal size in linear space, in the

$[n_s, 10\,n_s] \times [0, 140\,\mathrm{MeV}]$ region of the number density-temperature $(n_b, T)$ plane, which captures the whole domain of quark production. To do the counting we evaluate the quantity $N_{\mathrm{quark}} = \sum_j V_j Y_{\mathrm{quark},j} n_{b,j}$ separately for each bin, where the sum is performed over all the grid-cells $j$ of volume $V_j$ on a fixed time-slice in the orbital plane, and then normalized by the maximal count on the whole grid - where the maximal $N_{\mathrm{quark}}$ is marked with a star. The first row shows HQs at $t - t_{\mathrm{mer}} = 1.4\,\mathrm{ms}$, which span temperatures $\approx 40 - 60\,\mathrm{MeV}$ and densities $\approx 1 - 2\,n_s$. The second row shows WQs at $t - t_{\mathrm{mer}} = 3.3\,\mathrm{ms}$ produced at temperatures $\approx 10 - 30\,\mathrm{MeV}$ and densities $\approx 2 - 3\,n_s$. Finally, the third row for $t - t_{\mathrm{mer}} = 5.6\,\mathrm{ms}$ clearly shows that significant amounts of WQs and CQs can be present simultaneously. However, from the the colour code of this plot one can also see that the majority of quarks (indicated in yellow) are cold and dense.

## V. GRAVITATIONAL WAVE ANALYSIS

In this section we analyse the gravitational waveforms and their spectral properties. We use the Newman-Penrose formalism [78, 79] to relate the Weyl curvature scalar $\psi_4$ to the second time derivative of the polarization amplitudes of the gravitational wave strain $h_{+,\times}$ via

$$\ddot{h}_+ + i\ddot{h}_\times = \psi_4 := \sum_{\ell=2}^{\infty} \sum_{m=-\ell}^{m=\ell} \psi_4^{\ell,m} {}_{-2}Y_{\ell,m}, \qquad (1)$$

where ${}_sY_{\ell,m}(\theta, \phi)$ are spin-weighted spherical harmonics of weight $s = -2$. From our simulations we extract the modes $\psi_4^{\ell,m}$ with a sampling rate of $\approx 26\,\mathrm{kHz}$ from a spherical surface with $\approx 440\,\mathrm{km}$ radius centred at the origin of our computational domain and extrapolate the result to the estimated luminosity distance of 40 Mpc of the GW170817 event [28]. In

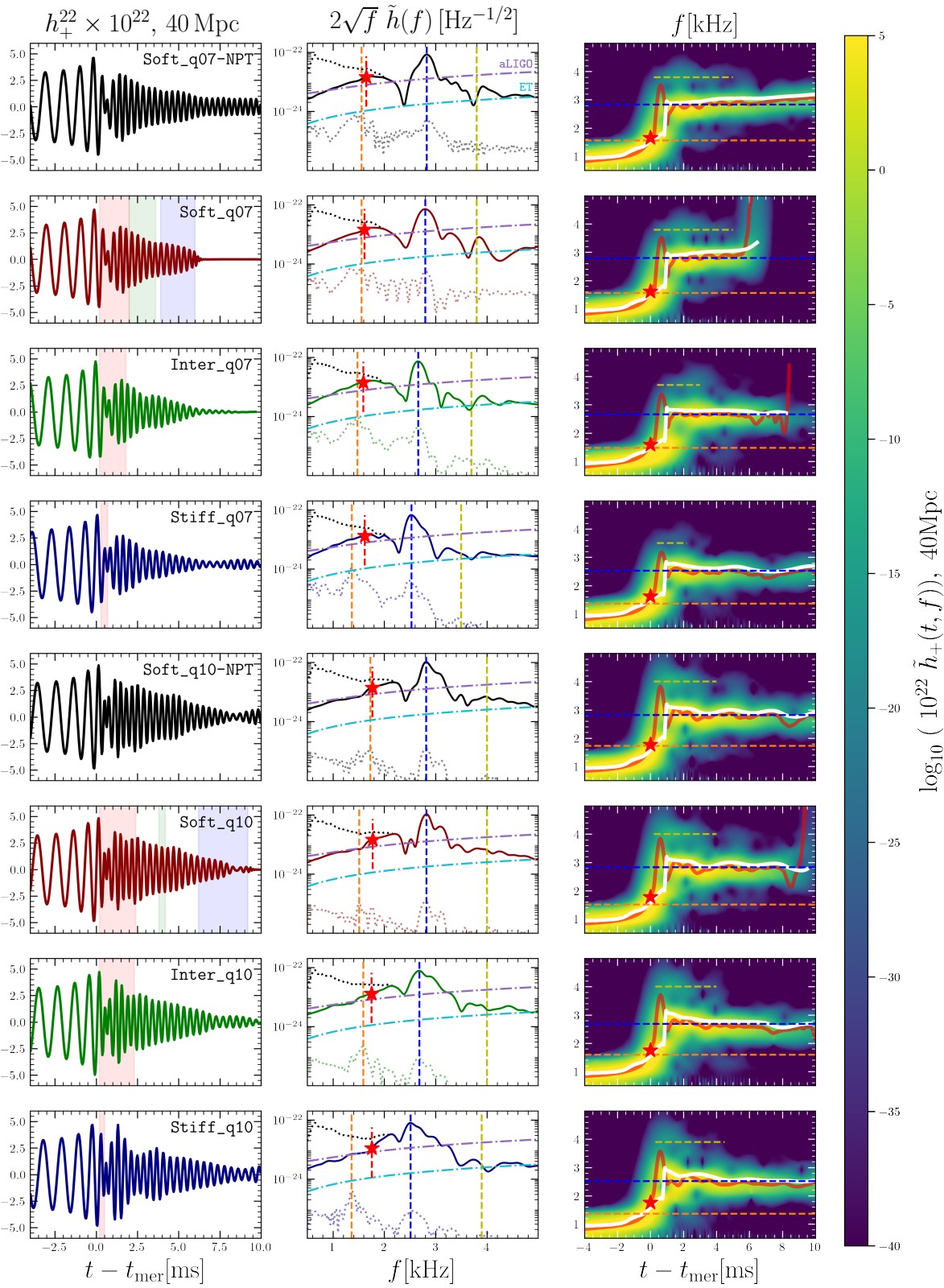

FIG. 5. Shown above is the $h_+^{22}$ of the strain, the power-spectral density of the effective strain, and the spectrogram of the $h_+^{22}$ of the `Soft`, `Inter`, and the `Stiff` EOSs considered in this work for $q = 1, 0.7$ and $\mathcal{M}_{\rm chirp} = 1.188$. Also included are the related spectra for `Soft-NPT` which does not include a phase transition to quark matter. In the right two panels the dashed blue and orange lines correspond to the $f^{2,2}$ and $f^{2,1}$ peaks respectively, the dashed yellow lines corresponds to the $f_3$ peak as measured in the spectrogram, and the star denotes the peak merger frequency, $f_{\rm mer}$. In the right figure, the white line traces the maximum in the spectrogram. Finally, the sensitivity curves in the middle plot are related the current sensitivity of advanced LIGO and the Einstein Telescope respectively [76, 77].

addition, we fix the angular dependence of the spherical harmonics part by the viewing angle $\theta = 15°$ determined from the jet of GW170817 [80] and set $\phi = 0°$ without loss of generality. We restrict our analysis to the dominant $\ell = m = 2$ and sub-dominant $\ell = 2, m = 1$ modes of the expansion (1). To analyze the spectral features of the waveforms we follow [81] and compute the power spectral density (PSD) defined as

$$\tilde{h}^{\ell,m}(f) := \frac{1}{\sqrt{2}} \left( \left| \int dt\, e^{-2\pi i f t} h_+^{\ell,m}(t) \right|^2 + \left| \int dt\, e^{-2\pi i f t} h_\times^{\ell,m}(t) \right|^2 \right)^{1/2}, \quad (2)$$

where the time integration is performed over the interval $t - t_{\mathrm{mer}} = 0 - 10\,\mathrm{ms}$ or up to the collapse to a black hole as is the case for the `Soft` binaries. Furthermore, we use spectrograms to study the time dependence of the gravitational wave frequency distribution. To perform the Fourier transform for the spectrograms we use time-windows of $3\,\mathrm{ms}$ centred at every $\approx 0.04\,\mathrm{ms}$ of our waveform data. We also study the phase difference between the $h_+^{2,2}$ and $h_\times^{2,2}$ modes by computing the instantaneous gravitational wave frequency $f_{\mathrm{GW}}$ defined as

$$f_{\mathrm{GW}} := \frac{1}{2\pi} \frac{d\phi}{dt}, \qquad \phi := \arctan\left( \frac{h_\times^{2,2}}{h_+^{2,2}} \right). \quad (3)$$

Finally, we can measure from $f_{\mathrm{GW}}(t)$ the instantaneous frequency at merger time defined as $f_{\mathrm{mer}} = f_{\mathrm{GW}}(t = 0)$.

In Fig. 5 we include a summary of our gravitational wave analysis of the configurations described in Tab. I. From top to bottom we show results for the different EOS models and the two mass ratios we have simulated. In the first column we show the dominant $h_+^{2,2}$ gravitational wave strain component with the time periods where the dominant quark production mechanism, HQ, WQ and CQ, are highlighted in red, green and blue, respectively.

The most prominent signature of quark matter in the waveform is clearly the earlier termination of the signal due to a phase-transition-triggered collapse to a black hole. This happens in our simulations only for the `Soft` model, which for both mass ratios forms a CQ stage that leads to a collapse of the HMNS. We note that models that include hyperons [82, 83] can lead to signatures similar as those from a first order deconfinement phase transition [3]. In both cases a softening of the EOS leads to more compact merger remnants that is closer to collapse. We estimate the lifetime $t_{\mathrm{BH}}$ of our merger remnants by the time at which the strain amplitude starts to decay exponentially as characteristic for the black-hole ring-down. In cases where no horizon is formed during simulation time, we provide lower limits on $t_{\mathrm{BH}}$ by the final time of the simulation. The values for $t_{\mathrm{BH}}$ we obtain in this way are listed in Table I. Comparing the waveforms of the `Soft` model for $q = 0.7$ (second row) and $q = 1$ (sixth row) shows that the unequal-mass case collapses $\approx 4\,\mathrm{ms}$ earlier than the equal mass case. This behaviour is consistent with the threshold mass analysis [84–86], which finds that highly asymmetric mergers tend to collapse at lower total mass than symmetric ones.

In the second column of Fig. 5 we show the PSDs computed via Eq. (2). Solid lines represent the dominant $\ell = m = 2$ mode, while light dotted lines are the sub-dominant $\ell = 2, m = 1$ mode. The dotted black line is the contribution to the PSD that comes from the inspiral part ($t - t_{\mathrm{mer}} < 0$), which is not included in the other curves. In addition we show estimated sensitivities of the advanced LIGO [76] (purple dot-dashed) and the Einstein Telescope [77] (cyan dot-dashed line). The PSD shows the typical three-peak structure from which a number of characteristic post merger frequencies can be extracted. The most prominent peak $f^{2,2}$ (dashed blue) corresponds to the dominant post-merger frequency of the $\ell = m = 2$ mode of the signal. Comparing the three EOS models for $q = 1, 0.7$ we find good agreement amongst their values of $f^{2,2}$ and only a small shift to lower frequencies with increasing stiffness of the EOS. Most notably, the $f^{2,2}$ peaks for the `Soft_q10` and `Soft_q10-NPT` are equivalent though the peak-width for `Soft_q10-NPT` is slightly broader (where the `NPT` simulations do not include a transition to quark matter as we will discuss below). In addition, we mark the maximal frequency $f^{2,1}$ of the sub-dominant $\ell = 2, m = 1$ mode by a dashed orange line and the merger frequency $f_{\mathrm{mer}}$ by a red star. Similar to the dominant mode, the $f^{2,1}$ frequencies (dashed orange) decrease only slightly with the increase in stiffness, but there is an appreciable difference when measuring the $f^{2,1}$ for the `Soft` EOS with and without a phase transition to quark matter in the equal-mass binaries. Overall we find that the sub-dominant mode in the equal mass case is by at least one order of magnitude suppressed with respect to the unequal-mass case. This is expected, since unequal-mass binaries result in less axially symmetric remnants than equal-mass binaries and, therefore, produce a richer post-merger frequency spectrum. In this sense gravitational waves from asymmetric binaries potentially provide more information about the EOS than symmetric binaries. However, in both cases the amplitude of the sub-dominant mode is orders of magnitude too low to be seen by current detectors, even when assuming the advantageous luminosity distance and sky localization of GW170817.

In the third column of Fig. 5 we show spectrograms where the colours from dark blue to light yellow indicate the distribution of low and high spectral weight during the evolution. In addition, we show lines for various characteristic frequencies: dashed blue and orange lines mark the $f^{2,2}$ and $f^{2,1}$ respectively - as extracted from the PSD. The dashed yellow line marks $f_3$ which we extract directly from the spectrogram because it is difficult to obtain unambiguously from the PSD. Finally, the solid red line is $f_{\mathrm{GW}}$ as computed from (3) and the solid white line follows the maximum values of the spectrogram. Note that the values of $f^{2,2}$ and $f^{2,1}$ extracted from the PSD align exceptionally well with the $f_1$ and $f_2$ peaks seen in the spectrogram; however, the $\ell, m = (2,2), (2,1)$ PSDs do not provide identifying information related to the $f_3$ peak which is measured directly from the spectrogram.

## A. Collapse Behaviour and Lifetime

To isolate the impact of quark matter on the lifetime we also perform simulations where we exclude the thermodynamically preferred quark matter phase at high temperature and densities in the `Soft` V-QCD model (see first and fifth row in Fig. 5) which we designate as the `Soft-NPT` model. Unlike the case with quark matter, the equal-mass case without quark matter does not collapse during our simulation time, however, we find that in the unequal-mass case, a binary using `Soft-NPT` model collapses after $\approx 11$ ms, i.e., roughly 5 ms later than with the `Soft` model. The early collapse of our simulations with the `Soft` V-QCD model point to a strong tension with the expected second-long lifetime of GW170817 [73, 74]. This means our results disfavour the `Soft` EOS and, therefore, allow us to further constrain the V-QCD model with information gained from the post-merger phase. This is an important result since, as discussed in Sec. II, the cold part of the `Soft` V-QCD model is fully consistent with currently known observations and the constraints from the post-merger lifetime can only be inferred from merger simulations.

Furthermore, obtaining faithful estimates for the lifetime of merger remnants that survive significantly longer than $\approx 10$ ms is difficult in our setup. There are mainly two reasons for this: First of all, resolving the formation of quark matter requires relatively fine resolution, which makes long simulation times numerically extremely expensive. We therefore restrict our high-resolution simulations to $\approx 10$ ms after merger. However, we present in Appendix B simulation results with medium resolution up to $> 35$ ms for the `Inter` and `Stiff` model. This allows us to approximate the lower bound of their lifetime to be $> 35$ ms. Second of all, the competition between turbulent heating in the presence of magnetic fields and cooling effects by neutrino emission can become important on time-scales on the order of several 100 ms [87]. Since we do not include either of these effects in our simulations we expect the thermodynamic properties, in particular the temperature distribution inside the star, not to be reliable on such long time-scales. However, we expect these thermal effects to be negligible on the comparably short time-scales the life-time estimates listed in Table I are based on.

## VI. SUMMARY AND CONCLUSION

We have studied fully 3+1 dimensional GRHD simulations of GW170817-like events using a unified model for QCD that includes the deconfinement phase transition from baryonic to quark matter. The focus of our work was the formation of quark matter during and after the merger of the two neutron stars. For this we studied three variants of the unified model, namely a soft, an intermediate and a stiff version. We identified three distinct mechanisms in the post-merger evolution during which mixed baryonic and quark matter and pure quark matter is formed and which we denote as: i) hot quark (HQ), ii) warm quark (WQ) and iii) cold quark (CQ) production. We find that some or all of these stages may be present in the

evolution depending on the masses of the stars and the EOS model employed.

Specifically, the HQ production appears at merger time and we find it to last for approximately two milliseconds, except for the stiff model where only a small amount of quarks are produced at merger time. During this time, strong shocks that are formed during the initial contact of the two stars lead to a steep rise in temperature. This causes a local formation of quark matter in the hottest, but not in the dense regions of the merger remnant. As the remnant begins to cool and lose its differential rotation causing an increase in density, the production of WQs may be possible depending on the model and binary configuration. The WQ production is the result of complicated merger dynamics and is observed in regions that are typically neither the hottest nor the densest regions of the HMNS. In this work WQs are only observed in the soft model. It is important to note that the existence and starting time of the WQ stage depends strongly on the details of the complicated post-merger evolution during which the HMNS pulsates in a highly non-axisymmetric way and, as such, turns out to be particularly sensitive to the mass ratio of the binary system. Additionally, we find that WQs are produced earlier and also in a larger quantities in non-equal mass binaries than in equal-mass binaries. Finally, the CQ production appears after the most violent post-merger period has settled and the density in the centre increases and cools down. The production of CQ is then formed in the dense and cold centre of the merger remnant. In this work, the CQ stage is followed by a phase-transition-triggered collapse to a black hole, using the terminology of [8]. The reason for this is the large latent heat of the deconfinement phase transition in the V-QCD model at low temperatures which does not allow stable cold quark matter cores inside isolated stars or HMNSs.

Finally, we studied the waveforms and their spectral properties and find the most significant imprint is the early termination of the gravitational wave signal due to the phase-transition-triggered collapse due to CQ production. By this we find the short post-merger lifetime of $\lesssim 10$ ms obtained from the soft version of V-QCD to be in tension with the expected second long lifetime of GW170817. It is remarkable that, even though the majority of the post-merger period ($\sim 6 ms$) of the unequal-mass case for the soft model includes a phase transition to a mixed or pure quark phase, there is essentially no measurable imprint on the amplitude or frequency of the gravitational wave strain. Moreover, this tension also means that there are new constraints for the predictions for the QCD equation of state arising from the merger simulations. As the soft version of the unified model is basically excluded by the lifetime estimate of the remnant, the soft versions should be also excluded from the analysis of the cold [29, 31] and hot [10] equation of state. We do not attempt to make such constraints precise in this article, but as an example we discuss the location of the critical point of the nuclear to quark matter transition. It was found in [10] that the temperature $T_c$ of the critical point varies roughly between 110 MeV and 130 MeV, with the soft version predicting the highest temperature. Exclusion of the soft model therefore means that $T_c \lesssim 120$ MeV within the framework.

There are multiple directions we plan to explore in future work. In the present work we only considered non-spinning binaries with individual masses consistent with GW170817, which prefers the low-spin prior. It will be important to investigate the dependence of various quark matter production stages also on different individual masses and spins of the binary components. Another direction worth studying is the impact of the deconfinement phase transition on the threshold mass, which has been analysed extensively in [84–86, 88] for purely hadronic EOSs as well as some hybrid models that include first order phase transitions[85, 86, 88]. Notice also that our simulations did not take into account the surface tension at the interface between the nuclear and quark matter. The value of the surface tension has not yet been computed in the V-QCD framework, and even if the value was known, it is not straightforward to take it into account: strong first order phase transitions in the presence of a sizeable surface tension proceed through bubble nucleation, which is a dynamical process that cannot be easily taken into account at the level of the equation of state. It is however clear that the impact of the surface tension would be to somewhat reduce the amount of mixed phase and suppress the quark production. We plan to study in a future work how large this effect is. Furthermore, it would be interesting to include cooling effects from neutrino emission which could have non-trivial impact on the quark formation, especially during the HQ and WQ stages. Finally, because of the high computational costs we restricted our high-resolution (medium-resolution) simulations to $\approx 10\,\mathrm{ms}$ ($\approx 40\,\mathrm{ms}$) after the merger. This allowed us to obtain reliable predictions for the quark matter production and the lifetime from the simulations that collapse during this short post-merger period. However, we can only speculate about the ultimate fate of the cases that do not collapse during our simulation time. We plan to extend these simulation to the order of 1 second by imposing axial symmetry in future long-term evolutions to obtain improved estimates for the lifetime of our long-lived HMNSs.

## ACKNOWLEDGMENTS

CE acknowledges support by the Deutsche Forschungsgemeinschaft (DFG, German Research Foundation) through the CRC-TR 211 'Strong-interaction matter under extreme conditions'– project number 315477589 – TRR 211. ST, KT, and LR acknowledge the support by the State of Hesse within the Research Cluster ELEMENTS (Project ID 500/10.006). TD and MJ have been supported by an appointment to the JRG Program at the APCTP through the Science and Technology Promotion Fund and Lottery Fund of the Korean Government. TD and MJ have also been supported by the Korean Local Governments – Gyeongsangbuk-do Province and Pohang City – and by the National Research Foundation of Korea (NRF) funded by the Korean government (MSIT) (grant number 2021R1A2C1010834). LR acknowledges funding by the ERC Advanced Grant "JETSET: Launching, propagation and emission of relativistic jets from binary mergers and across mass scales" (Grant No. 884631). The simulations were performed on HPE Apollo HAWK at the High Performance Computing Center Stuttgart (HLRS) under the grant BNSMIC. For our visualisations we made an extensive use of the Kuibit library [89].

## Appendix A: Convergence Analysis

To ensure a high level of confidence in the discussed results, we have performed a consistency study with the Soft_q10 initial data to ensure that the overall dynamics of the merger and the phase transition-induced collapse are not simply artefacts of the numerical accuracy. To this end, we have employed three resolutions which are characterised by the finest grid spacing of $\Delta_\mathrm{L} := 369\,\mathrm{m}$, $\Delta_\mathrm{M} := 295\,\mathrm{m}$ and $\Delta_\mathrm{H} := 221\,\mathrm{m}$ where $\Delta_\mathrm{H}$ is the resolution used for the results presented in the main body of the text. Additionally, we employ a Courant static time step such that $dt = C_{dt}\max(dx)$ where $C_{dt} = 0.2$ for our simulations. Therefore, an increase in spatial resolution results in an increase in temporal resolution.

As shown in Fig. 6 (left) the gravitational waveforms show only a slight variation in the relative phase during the inspiral and a systematic decrease in the post-merger amplitude, however, all resolutions considered result in the same collapse time. Furthermore, we see in Fig. 6 (right) that the existence and identification of the different stages of quark production is robust under the change of resolution. Notably, the intermediate production of WQ appears for $\Delta_\mathrm{H}$ which is not surprisingly absent in the $\Delta_\mathrm{M}$ resolution given the sensitivity required in temperature and density for WQ to appear especially for such a short duration ($\sim 0.3\,\mathrm{ms}$). Remarkably, the peaks in $Y_\mathrm{quark}$ follow a similar spacing in time, where individual peaks tend to form more often with an increase in resolution.

## Appendix B: Medium Resolution Runs

Given the consistent nature of our evolutions as discussed in A, we conducted additional evolutions at $\Delta_\mathrm{M}$ resolution up to $\approx 35\,\mathrm{ms}$ for the Inter and Stiff models. The results of these runs are summarised in Fig. 7 where the behaviour of the thermodynamic quantities is shown to be consistent with the results in Fig. 2 as well as the waveforms shown in Fig. 5. This allows us to place lower bounds on the collapse time, namely $\approx 35\,\mathrm{ms}$. Although these results provide evidence that the Inter and Stiff models are consistent with the expected second long lifetime of GW170817, we want to emphasize that our simulations do not include contributions of magnetic fields and neutrino cooling which could be relevant for the long-term stability of a HMNS.

## Appendix C: Extraction Radii of Thermodynamic Quantities

In our analysis of the quark production within the remnant HMNS it seemed pertinent to not only discuss the amount of quarks produced, but also the relative location of quark production. To illustrate the extraction radii we present in Fig. 8

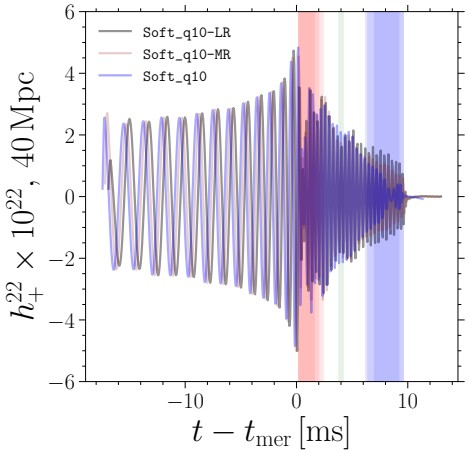 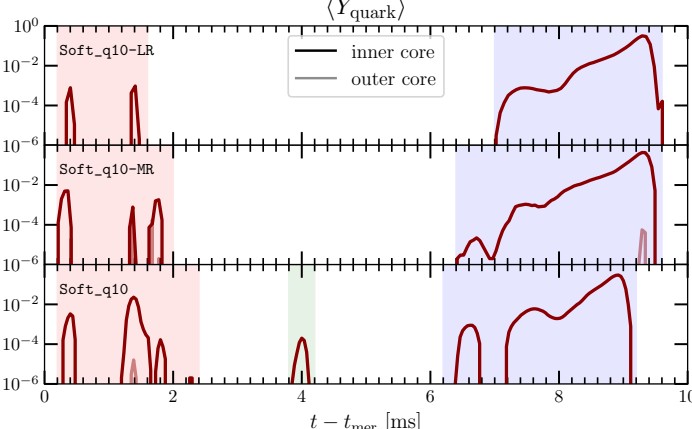

FIG. 6. Shown are the consistency results of the `Soft_q10` dataset. Left: consistency as measured in the gravitational wave strain $h_+^{22}$ polarisation amplitude. Although slight changes in the amplitude and phase are observed, the collapse time is the same in all resolutions. Right: the corresponding quark fraction for the three resolutions considered. Most notable is the appearance of WQ in the high-resolution run further emphasising the need for high resolutions to capture fine-structure details.

these regions for the `Inter_q10` run, for which there is no collapse to a black hole on the timescale that we cover. Lime contours within the figure correspond to baryon-number density $n_b$ equal to $1n_s$, $2n_s$ and $3n_s$. The regions enclosed by the black dotted circles correspond to radii of 3 km, 6 km and 9 km which we designate as the inner core, outer core and inner crust. The annuli are centred on the maximal density point for $t < t_{mer}$ and on the centre of mass of the system for $t \geq t_{mer}$.

These extraction domains are then used to extract the local average and maximal values of thermodynamic quantities as shown in Fig. 2 and Fig. 7. The averages over any annuli $\Omega$ are obtained using the following formula:

$$\langle f \rangle := \frac{1}{A} \int_\Omega f d^2x, \qquad A := \int_\Omega d^2x, \qquad \text{(C1)}$$

where, for simplicity, we consider the metric determinant of flat spacetime in the integrand.

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

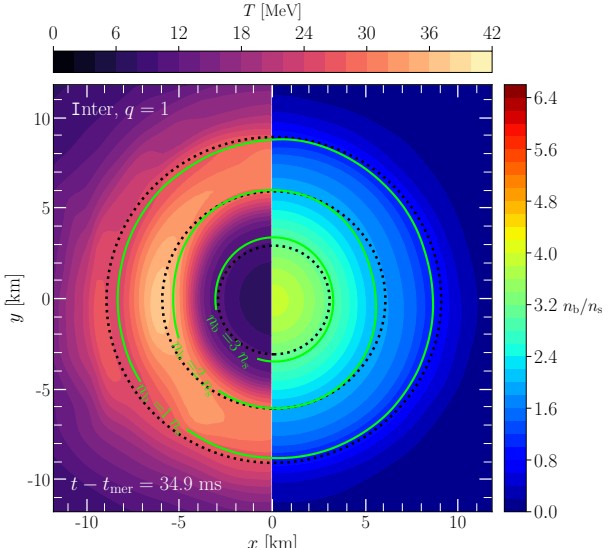

FIG. 8. A slice on $(x,y)$ plane from the `Inter_q10-MR` run, plotted at the latest available time. Shown are the temperature (left part) and the baryon-number density (right part), while the black dashed lines represent the 3, 6, and 9 km ranges over which the integrals are computed. Note the good match with the normalised baryon-number $n_{\rm b}/n_s$ at values of 3, 2, and 1, respectively (green solid isocontours).

[hep-th].

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
