# Peer review of "Quark formation and phenomenology in binary neutron-star mergers using V-QCD"

_SciPost Physics_

## Round 1 · Referee Report · Anonymous (Referee 1) · 2022-7-6

Report

The authors explore channels of quark formation in the context of binary neutron star (BNS) mergers by employing a very reasonable set of equations of state (EOSs) that are derived from a novel framework of V-QCD and are compatible with contemporary pulsar observations. To this aim, the authors chronologically classify stages of postmerger evolution that show an appearance of quark matter dictated by the temperatures and densities probed by the mergers. In addition, the authors compute gravitational wave (GW) signatures of such a phase transition and explore the parameter space of the V-QCD model by considering EOSs of different stiffnesses. The most robust signature of phase transitions is found to be an early time of collapse for softer EOSs which is consistent with previously published numerical studies.

In my opinion, the paper could be published after a satisfactory addressing of the following concerns/alterations:

Note: [1] and [2] are the references provided at the end of the report.

  • In the abstract, the authors emphasize 'multi messenger' observations as constraining extreme density regions of the EOS however, they focus exclusively on GW signatures and use the collapse time to rule out the soft EOS. The abstract thus needs to be modified.

  • The fact that the spatial distribution of quark matter strongly correlates with local regions of high temperature and that there are crossings into a mixed quark phase at low and high temperatures has been observed in [2] and this work should be cited in the conclusions.

Additionally, in an asymmetric merger, spatial distributions of quark matter are also asymmetric is a point made explicit in both [1] and [2]. These works should be also be referenced from the conclusions.

  • The GW amplitude is the strongest at the merger and the 1st few ms after merger which correspond to the hot quark regime. During these times, the metric is at its most dynamical. It is thus not reasonable to ignore the square root of the determinant of the 3-metric in a calculation of quark fraction in post precessing at least for the hot-quark regime. What are the reasons to ignore the volume form ?

  • For a given postmerger time instant, are the extraction radii of 3, 6 and 9 km for the calculation of quark fraction really representative of all the quarks appearing in the x-y plane? Very shortly after the merger, the remnant is highly deformed and it is difficult to distinguish an inner core from an outer core, less so in terms of circular radii.

It will be very convincing to see a diagnostic computation of average quark fraction (for a representative binary say the q=1 soft EOS) across the entire x-y plane or better yet, the entire volume and a plot of this quantity as a function of time from merger while retaining the volume form of the metric. Does this time variation also show regimes of hot/warm/cold quarks ?

Has the correspondence of annular regions of 3, 6 and 9km with 1ns, 2ns, 3ns density contours as shown in Figure 8 been verified for asymmetric systems and soft/stiff EOSs?

  • Do the boundaries between hot and warm quarks as shown in the EOS parameter space in Figure 4 change if we compute <Y_quark> for the entire volume with curvature corrections ?

  • Regarding Section IV , Paragraph 5, '... a precise definition of the warm quark stage is difficult since it sensitively depends on the combination of densities and temperatures ...' , are these postmerger stages of hot, warm and cold quarks sufficiently general and not an artefact of the V-QCD model?

A discussion on the generality of these regimes in the conclusions will carry merit.

  • The claim that the remnant for GW170817 survived for about a second is based on simulations of jet structure and kilonova emissions. No GW postmerger was observed for this event. In the absence of such a signal, one should not entirely rule out the models of the V-QCD EOSs which predict shorter-lived remnants. Would the authors please provide a comment?

The authors remark in the conclusions that '... there is essentially no measurable imprint on the amplitude or frequency ...' but go on to say that due to '(this tension between postmerger lifetimes) there are new constraints for the predictions for the QCD EOSs ...'. A brief clarification and a qualitative discussion on what these new constraints are should be added to the conclusions.

Refrences:

[1] E. R. Most, L. J. Papenfort, V. Dexheimer, M. Hanauske, S. Schramm, H. Sto ̈cker, and L. Rezzolla, Signatures of quark-hadron phase transitions in general- relativistic neutron-star mergers, Phys. Rev. Lett. 122, 061101 (2019), arXiv:1807.03684 [astro-ph.HE].

[2] A. Prakash, D. Radice, D. Logoteta, A. Perego, V. Ne- dora, I. Bombaci, R. Kashyap, S. Bernuzzi, and A. Endrizzi, Signatures of deconfined quark phases in binary neutron star mergers, Phys. Rev. D 104, 083029 (2021), arXiv:2106:07885[v2] [astro-ph.HE].

PS: Please note a minute error in the last point of the original report. The statement should be '... which predict shorter lived remnants ' instead of 'longer lived remnants'.

  • validity: top
  • significance: good
  • originality: good
  • clarity: good
  • formatting: excellent
  • grammar: excellent

Author:  Samuel Tootle  on 2022-08-10  [id 2719]

(in reply to Report 2 on 2022-07-06)
Category:
answer to question
correction

Dear referee,

We would like to thank you for handling our paper so quickly and for the excellent comments which will certainly lead to an improvement of our manuscript. In the following we address all the points raised by both referees and summarize our revisions on the manuscript.

  • In the abstract, the authors emphasize 'multi messenger' observations as constraining extreme density regions of the EOS however, they focus exclusively on GW signatures and use the collapse time to rule out the soft EOS. The abstract thus needs to be modified.

This is correct and we thank the referee for spotting this discrepancy.
The revised abstract now correctly focuses purely on the GW observables.

  • The fact that the spatial distribution of quark matter strongly correlates with local regions of high temperature and that there are crossings into a mixed quark phase at low and high temperatures has been observed in [2] and this work should be cited in the conclusions.

We are grateful to the referee for highlighting that the behavior of the quark matter distribution as a function of temperature and density has been described in previous works.
We have added a citation to reference [2] in the second paragraph of our Conclusion to emphasize previous findings.

Additionally, in an asymmetric merger, spatial distributions of quark matter are also asymmetric is a point made explicit in both [1] and [2]. These works should be also be referenced from the conclusions.

Indeed, the impact of asymmetric binary initial data on the quark matter distribution has been studied in [1,2]. We thank the referee for highlighting this point which further validates our results by properly acknowledging previous works. We have added a corresponding statement where we cite these two references in the third paragraph of our Conclusion.

  • The GW amplitude is the strongest at the merger and the 1st few ms after merger which correspond to the hot quark regime. During these times, the metric is at its most dynamical. It is thus not reasonable to ignore the square root of the determinant of the 3-metric in a calculation of quark fraction in post precessing at least for the hot-quark regime. What are the reasons to ignore the volume form ?

We thank the referee for their comment related to the absence of the volume form in the calculation of the quark fraction. To ascertain the influence of the volume form on the integral, we have carried out additional simulations for both of the soft EOS configurations where the metric determinant has been properly stored for use in post-processing.

To this end we find, maybe somewhat surprisingly, that the computed volume averages are robust as compared to those computed with sqrt(det(g)) except right before black hole formation. We have checked that in the averaging procedure, the square root of the determinant of the 3-metric can be neglected with the ~10-20% relative difference occurring only at the onset of collapse. We have added a plot to the reply (Yquark_avg_detg_vs_flat_with_reldiff.pdf) that shows the result of this comparison and added a corresponding statement in Appendix C.

  • For a given postmerger time instant, are the extraction radii of 3, 6 and 9 km for the calculation of quark fraction really representative of all the quarks appearing in the x-y plane? Very shortly after the merger, the remnant is highly deformed and it is difficult to distinguish an inner core from an outer core, less so in terms of circular radii. It will be very convincing to see a diagnostic computation of average quark fraction (for a representative binary say the q=1 soft EOS) across the entire x-y plane or better yet, the entire volume and a plot of this quantity as a function of time from merger while retaining the volume form of the metric. Does this time variation also show regimes of hot/warm/cold quarks ? Has the correspondence of annular regions of 3, 6 and 9km with 1ns, 2ns, 3ns density contours as shown in Figure 8 been verified for asymmetric systems and soft/stiff EOSs?

We thank the referee for this comment. We have checked that the annuli regions, which extend up to 9 km, cover all instances of quark formation in the orbital plane. Furthermore, the designation of the three radii have been chosen in an effort to highlight characteristic times and regions of quark production.
As the referee correctly identified, right after the merger, when the remnant is highly deformed, the anular regions do not closely follow isodensity contours in the x-y plane. However, the defined regions serve to provide a qualitative understanding of average quark formation in different temperature and density regimes. We note below and in our revised conclusion that we intend in future works to perform 3D analysis of the quark formation and distribution once our codes have been appropriately modified to make this process efficient. Finally, we have added to this reply a plot that shows the time series of the averaged quark fraction in the x-y plane (including the metric determinant) over this region.

Additionally, the naming convention of "inner/outer core" and "inner crust" have been motivated by the structure of the soft q=1 remnant at late times, however these definitions serve as consistent reference labels in the text even during the violent merger process when these descriptors are not accurate. We agree that right after the merger and for asymmetric configurations this division into regions does not have a clear physical motivation, however, they do provide a useful qualitative description and insight into the studied quark formation stages.

  • Do the boundaries between hot and warm quarks as shown in the EOS parameter space in Figure 4 change if we compute <Y_quark> for the entire volume with curvature corrections ?

The referee is correct. This further motives the need for our follow-on effort to analyse the 3D distribution of the quark fraction and the corresponding volume averages which could indeed lead to new and interesting insights. In our exploratory work, we present our initial findings which have not only provided insights into the reported quark formation stages, but also given us vital information that will enable a more detailed analysis to include the 3D distribution of the quark fraction and the corresponding volume averages. This, however, was beyond the current scope of this project as we did not know a priori the time periods that could be interest for analyzing quark formation nor the frequency for which data would need to be captured for 3D analysis. Storing and processing 3D requires not only computational and storage resources, but also proper planning in order for efficient analysis of such large datasets. Additionally, we currently compute the quark fraction as a post-processing step using the EOS table which further adds to the complexity when handling 3D data. As such, we are currently improving our code to enable on-going work to compute and store quark matter related quantities during the simulation thereby allowing us to properly analyze the three-dimensional distribution of quark matter with the inclusion of spacetime curvature.

  • Regarding Section IV , Paragraph 5, '... a precise definition of the warm quark stage is difficult since it sensitively depends on the combination of densities and temperatures ...' , are these postmerger stages of hot, warm and cold quarks sufficiently general and not an artefact of the V-QCD model?

    A discussion on the generality of these regimes in the conclusions will carry merit.

This is an important point. In order to analyze with certainty how model dependent our results are would require running the simulations for a significantly wider class of EOSs and constituent masses. However, the results depend on the V-QCD model only through the equation of state, which is featureless in the relevant regime apart from the location and strength of the phase transition. The variation of these parameters in the range allowed by the V-QCD model is represented by the soft, intermediate and stiff variants, but since we are focusing on simulations at the chirp mass of GW170817 our results for the different EOSs are quite different apparently due to the different threshold mass for collapse into a BH. However, nothing in our simulation results suggests that the stages of quark production would be strongly dependent on the EOS. To the contrary, the details of the simulations (in Figs. 2 and 3) suggest that the production is driven by merger dynamics, and happens whenever the state of the matter reaches the phase transition line. Simulations with EOSs having significantly weaker phase transition might lead to a slightly modified picture, for example, the cooling of the quark matter in the final "cold quarks" might be weaker, but such a phase transition would contradict our EOS model. But we would say that this is a prediction of the model rather than an artefact.

In summary, for EOS models with phase boundaries in the density-temperature plane similar to the V-QCD model we expect that hot and warm quark stages appear in a similar manner. However, as we point out below (see Fig.3) there exist studies [75] (1912.09340) that predict a hot quark core right before collapse and, therefore, no cold quark stage. This has to do with the fact that [75] uses a simple parametrization of the temperature dependence in the quark phase that scales with density and is agnostic to the matter composition. This is different to V-QCD where the temperature dependence in the mixed (quark) phase is a result of thermodynamic calculation, which inherits the temperature dependence of the quark part from the underlying dual black hole description that is tuned to QCD quark matter and the baryonic part from the finite volume corrected and effective potential improved hadron resonance gas matched to cold V-QCD baryon EOS. In this sense our temperature dependence in the mixed and quark matter phase does contain information about the matter composition and, thus, the cold quark phase we find is a consequence of that. We have added a corresponding statement in the end of the first paragraph of the Conclusion section to highlight this unique outcome when using the V-QCD models.

  • The claim that the remnant for GW170817 survived for about a second is based on simulations of jet structure and kilonova emissions. No GW postmerger was observed for this event. In the absence of such a signal, one should not entirely rule out the models of the V-QCD EOSs which predict shorter-lived remnants. Would the authors please provide a comment?

    The authors remark in the conclusions that '... there is essentially no measurable imprint on the amplitude or frequency ...' but go on to say that due to '(this tension between postmerger lifetimes) there are new constraints for the predictions for the QCD EOSs ...'. A brief clarification and a qualitative discussion on what these new constraints are should be added to the conclusions.

We thank the referee for highlighting this point. As the referee correctly stated, the remnant lifetime of GW170817 is only estimated to have been on the order of 1s which has been inferred from the multi-messenger detections and the follow-on GRMHD simulations. We have since added clarifying remarks in the conclusion to this effect and emphasized that continued multi-messenger detections as well as increased sensitivity of future GW detectors which can provide post-merger waveforms will be necessary to provide constraints to the V-QCD model based on the lifetime inferred from the GW signal. We have rephrased the corresponding statements in the conclusion and point out that strictly ruling out the model would require a reliable post-merger waveform measurements.

Refrences:

[1] E. R. Most, L. J. Papenfort, V. Dexheimer, M. Hanauske, S. Schramm, H. Stöcker, and L. Rezzolla, Signatures of quark-hadron phase transitions in general- relativistic neutron-star mergers, Phys. Rev. Lett. 122, 061101 (2019), arXiv:1807.03684 [astro-ph.HE].

[2] A. Prakash, D. Radice, D. Logoteta, A. Perego, V. Nedora, I. Bombaci, R. Kashyap, S. Bernuzzi, and A. Endrizzi, Signatures of deconfined quark phases in binary neutron star mergers, Phys. Rev. D 104, 083029 (2021), arXiv:2106:07885[v2] [astro-ph.HE].

Attachment:

Yquark_avg_detg_vs_flat_with_reldiff.pdf

---

## Round 1 · Referee Report · Anonymous (Referee 2) · 2022-7-26

Strengths

The novel V-QCD equation(s) of state for Neutron Star matter is(are) combined with solid, state-of-art merger simulations. The results are very interesting and provide valuable insights for the field.

Weaknesses

All weaknesses inherent to the V-QCD model underlying the equation of state used as an input for the BNS merger simulations are mentioned and addressed satisfactorily.

Report

Warnings issued while processing user-supplied markup:

  • Inconsistency: Markdown and reStructuredText syntaxes are mixed. Markdown will be used.
    Add "#coerce:reST" or "#coerce:plain" as the first line of your text to force reStructuredText or no markup.
    You may also contact the helpdesk if the formatting is incorrect and you are unable to edit your text.

The authors perform state-of-the-art binary neutron star merger simulations as implied by the equation of state predicted within the V-QCD model. The latter combines information coming from holography in the so-called Veneziano limit, Lattice-QCD, nuclear models and chiral effective theory, and is compatible with recent astrophysical observations.

Quite interestingly, the analysis reveals three (mostly distinct) instances of quark matter formation -- which is absent for individual stars in equilibrium in this framework -- dubbed as Hot, Warm and Cold quark matter. They are argued to take place within patches of decreasing temperatures and increasing densities, occuring at different stages of the merger.

The GW spectrum resulting from this process are also studied, the goal being to find quark-formation-related imprints in the GW spectrum possibly observable in the near-future. The authors highlight the early collapse time as the main signature of these phase transitions. This also generates some tension between the results presented here for the softer version of the V-QCD EOS and the analysis of the GW170817 event in Refs.[73,74].

I believe this is a very interesting and well written preprint, which should be published. I do, however, have some brief questions/comments:

  • What do the authors mean by beta-equilibrated matter in this model? As far as I know, in the V-QCD model at hand baryonic matter comes from a homogeneous approximation loosely based on classical instanton solutions in the bulk, so there is no notion of isospin asymmetry nor matter built from protons and neutrons.

  • Temperature in the dense matter phase is included through a van der Waals model on top of the holographic framework. Besides the comparison with Ref.[75] related to the temperature of quark cores, do the authors have any comments/estimations on how dependent the results presented in this paper are on this choice?

  • The initial conditions are chosen to be somewhat similar to those associated to the GW170817 event. It might be interesting to include a more detailed comparison of the GW signals computed in this paper as compared to the former observations, that goes beyond the discussion of the collapse time given in Sec. V.A.

  • It is suggested in Sec.VI that including the surface tension and the bubble nucleation process that presumably drives the phase transition, which are ignored in the present setting, would suppress quark production. In contrast, is there any other phenomena that would be expected to enhance the quark formation and/or its signatures in, say, the gravitational wave spectrum?

  • The dependence of the results presented in this paper on the chosen spatial resolution is shortly addressed in App. A. Although the existence of the different stages of quark formation and the resulting collapse time seem to be robust, the more detailed account of the different simulations shows some appreciable differences, see Fig. 6 (right). It would be important to quantify this further.

  • validity: high
  • significance: high
  • originality: good
  • clarity: high
  • formatting: perfect
  • grammar: perfect

Author:  Samuel Tootle  on 2022-08-10  [id 2720]

(in reply to Report 3 on 2022-07-26)
Category:
answer to question

Dear referee,

We are grateful for the referee's time and attention to detail in the review of our manuscript and for the excellent comments which will certainly lead to an improvement of our manuscript. In the following we address all the points raised by the referee as well as the proposed changes in the manuscript.

  • What do the authors mean by beta-equilibrated matter in this model? As far as I know, in the V-QCD model at hand baryonic matter comes from a homogeneous approximation loosely based on classical instanton solutions in the bulk, so there is no notion of isospin asymmetry nor matter built from protons and neutrons.

We thank he referee for pointing this out. Indeed there is no dependence on the charge fraction from the V-QCD model computation. But as explained in Ref. [10], for the finite temperature model in the nuclear matter we use a van der Waals model of protons, neutrons and electrons which does give rise to charge/ electron fraction dependence and the notion of beta-equilibrium. However the prediction of the ven der Waals model turns out not to be viable: it gives too low symmetry energy. Therefore for the final model we simply use the prediction of HS(DD2) for the dependence on the electron fraction.

  • Temperature in the dense matter phase is included through a van der Waals model on top of the holographic framework. Besides the comparison with Ref.[75] related to the temperature of quark cores, do the authors have any comments/estimations on how dependent the results presented in this paper are on this choice?

We expect that the dependence of the results on this choice is weak. This is because the temperature dependence in the nuclear matter phase is presumably weak in the range of temperatures probed by the simulations -- it cannot be computed reliably from QCD but all model estimates predict weak dependence. We have also seen this in early low resolution simulations where we tested an EoS that borrowed the temperature dependence from HS(DD2) instead of the van der Waals model, and obtained very similar results as for the van der Waals construction.

  • The initial conditions are chosen to be somewhat similar to those associated to the GW170817 event. It might be interesting to include a more detailed comparison of the GW signals computed in this paper as compared to the former observations, that goes beyond the discussion of the collapse time given in Sec. V.A.

We thank the referee for this suggestion, however, even though a detectable chirp was obtained from the GW170817 event, a fully reconstructed waveform has not been made available to our knowledge. Furthermore, the more interesting aspects of the power spectral density such as the influence on the l=2,m=1 multi-pole mode is well below the noise floor of the current generation of detectors.

  • It is suggested in Sec.VI that including the surface tension and the bubble nucleation process That presumably drives the phase transition, which are ignored in the present setting, would Suppress quark production. In contrast, is there any other phenomena that would be expected To enhance the quark formation and/or its signatures in, say, the gravitational wave spectrum?

We thank the referee for raising this line of inquiry. To the best of our understanding, we are not aware of additional physical processes that would contribute to additional quark production. Furthermore, choosing a different model (e.g., something other than the soft, intermediate, and stiff utilized in this work) could be generated in such a way that the transition to quark matter is more easily obtained. However, such a model will likely result in a much softer EOS which could be outside of the current astrophysical constraints set by NICER and PSR observations.

  • The dependence of the results presented in this paper on the chosen spatial resolution is shortly addressed in App. A. Although the existence of the different stages of quark formation and the resulting collapse time seem to be robust, the more detailed account of the different simulations shows some appreciable differences, see Fig. 6 (right). It would be important to quantify this further.

We thank the referee for highlighting an important aspect that deserves further analysis with regards to quantifying the averaged quark production as a function of resolution. As we see in Fig. 6, the amount of quark matter produced for a given stage is fairly robust, however, with an increase in the evolution resolution the distribution of<Y_quark> in time and space are influenced thereby hinting that we may not yet be in a fully convergent regime with respect to quark production. This is one aspect of the research we intend to pursue in a future work, as we have mentioned in response to referee 2, where we intend to perform 3D analysis of the production and distribution of quark matter which will be crucial to fully quantify the resolution dependence in GRHD simulations.

---

## Round 1 · Referee Report · Ronnie Rodgers (Referee 3) · 2022-8-8

Strengths

1 - The holographic approach (V-QCD) used to obtain intermediate density equations of state is state-of-the-art, incorporating many considerations in order to match known physics as closely as possible.
2 - The authors perform sophisticated merger simulations, from which they extract concrete results, identifying multiple stages of quark formation and further constraining the holographic model.

Weaknesses

1 - I feel that some further discussion of the reliability and limitations of the holographic approach are warranted.

Report

The authors simulate binary neutron star mergers, using holography to provide model equations of state for nuclear and quark matter in the challenging cold, intermediate density regime. From the evolution of the quark fraction over the course of the mergers, the authors identify multiple stages of quark production, distinguished by the temperature and density at which they occur. The gravitational wave signals of the mergers are also studied. The softest equation of state used leads to rapid collapse and termination of the signal, in tension with observation.

I believe the article meets the expectations and acceptance criteria, and should thus be published. In particular, the work is a novel example of holographic phenomenology, and opens multiple clear directions for significant follow-up work. I do have some brief questions and requests for clarification:

1 - In figure 1, the chosen equations of state span quite a narrow range within the allowed V-QCD region, particularly at energy densities below around 500 MeV/fm\(^3\). Do the authors have any comments on the reason for this choice, and whether it has a significant impact on the results?

2 - In order to obtain more realistic, non-trivial temperature dependence of the equation of state in the nuclear matter phase, the authors use a van der Waals model that matches on to the V-QCD result at zero temperature \(T\). Meanwhile, in the quark matter phase the authors use V-QCD also at \(T > 0\). Can the authors provide any clarification on why they trust the the \(T\)-dependence from V-QCD in the quark matter phase, given that it gives the wrong answer in the nuclear matter phase?

3 - For the soft equation of state, simulations at multiple resolutions were performed to test convergence. Warm quark production was only seen at the highest resolution. Are the authors confident that the conditions for warm quark production are not met with intermediate or stiff equations of state, and would not be seen with yet higher resolution?

Requested changes

1 - It took me longer than necessary to understand the meaning of the squares, stars, and circles in the left panel of figure 4, since they are only very briefly mentioned in the text. Since these markers are important in explaining the different quark production phases (one of the key results in the paper), I think it would be worthwhile to add a description of the precise meaning of them, i.e. that they show the temperature and density at the point with maximum value of the corresponding observable.

  • validity: high
  • significance: good
  • originality: high
  • clarity: high
  • formatting: excellent
  • grammar: excellent

Author:  Samuel Tootle  on 2022-08-11  [id 2723]

(in reply to Report 4 by Ronnie Rodgers on 2022-08-08)
Category:
answer to question

We are grateful to Dr. Rodgers for his careful review of our work and the recommended required change to enable an easier review of Fig. 4 for the reader. We hope the following clarifying remarks answer Dr. Rodgers' questions and that the correction enables a clearer understanding when first reading Fig. 4.

1 - In figure 1, the chosen equations of state span quite a narrow range within the allowed V-QCD region, particularly at energy densities below around 500 MeV/fm

There is a simple explanation for this. First of all, the low density part, up to ~250MeV/fm^3 (around 1.5 nuclear saturation density),is described by a single nuclear theory model which captures the matter properties better than the holographic homogeneous baryon model which becomes unreliable at lower densities where the homogenous assumption is not justified. This essentially restricts the EOS in Fig 1 to a single line below 250MeV/fm^3. Second of all, for 250 up to 500 MeV/fm^3 (and beyond) the VQCD model is used. In principle the model there has quite some freedom indicated by the pink shaded area. However, the matching to the low density nuclear theory model and by demanding consistency with astrophysical observations (two-solar mass bound, tidal deformabilty by GW170817, radius constraints by the NICER experiments) results in a quite small variablity whose bounds are faithfully covered by the soft (red, lower bound) and the stiff (blue, upper bound) in the range the referee points out. The impact on the results is also easy to understand from Fig 1 (right), where we mark by various symbols the individual masses and the corresponding radii of the individual stars used to prepare the binary initial data for the merger simulations. By comparing the radii of different EOSs at fixed mass, the difference is of the order of a few 100 meters. As a consequence the inspiral parts of the waveforms depend only mildly on the EOS model. However, the situation is very different during and after merger where the densities reached are much higher than 500MeV/ fm^3 where the difference between the EOSs is much larger. Note that although the differences in pressure shown in Fig. 1 seem small this due to log scale utilized and, thus, the differences are actually sizable compared to low densities. This means the post-merger wavefoms and the ultimate fate of the remnant (black hole or HMNS) differ significantly.

2 - In order to obtain more realistic, non-trivial temperature dependence of the equation of state in the nuclear matter phase, the authors use a van der Waals model that matches on to the V-QCD result at zero temperature T. Meanwhile, in the quark matter phase the authors use V-QCD also at T > 0. Can the authors provide any clarification on why they trust the the T-dependence from V-QCD in the quark matter phase, given that it gives the wrong answer in the nuclear matter phase?

The referee is correct in that an in-depth analysis on the construction and nuances of the VQCD model are not covered in this work. Instead, we highlight the relevant aspects of the model to support the results of this work and we refer the interested reader to an in-depth introduction on how the EOS model is constructed which can be found in [10,40], as we point out in the first paragraph of Section II. The short answer to the referees question is that V-QCD baryons are defined on the gravity side on a singular background geometry (of the good kind), so there is no BH horizon which could provide non-trivial T-dependence in the confined phase. On the other hand the deconfined quark phase is based on a dual black hole geometry and the non-trivial T-dependence can be obtained from the usual BH thermodynamics. This is a consequence of the large Nc limit that can't be easily avoided and is reminiscent of the typical Hawking-Page phase transition in holographic constructions. However, T-dependence is essential to describe the complicated fluid dynamics in the post-merger stage. We found that using a vdW construction that inherits the behaviour of the cold V-QCD baryons was the simplest way to include T-dependence in this phase without making too many ad-hoc assumptions.

3 - For the soft equation of state, simulations at multiple resolutions were performed to test convergence. Warm quark production was only seen at the highest resolution. Are the authors confident that the conditions for warm quark production are not met with intermediate or stiff equations of state, and would not be seen with yet higher resolution?

We thank the referee for also highlighting this important, but also highly non-trivial question which has also been briefly discussed with Referee 1 and 2. As the referee points out, we find that resolving the warm quark stage indeed requires sufficiently high resolution. Based on this observation we have produced all results shown in the main text with the highest resolution considered such that we were able to ensure convergence in the gravitational wave properties and qualitatively similar physics in the production of mixed/ pure quark matter. In a follow-up work, we plan to explore in full 3D the distribution of quark matter in the star with a focus on attempting to ascertain whether convergence can be obtained in the production of quark matter when higher resolutions are used.

Requested Changes 1 - It took me longer than necessary to understand the meaning of the squares, stars, and circles in the left panel of figure 4, since they are only very briefly mentioned in the text. Since these markers are important in explaining the different quark production phases (one of the key results in the paper), I think it would be worthwhile to add a description of the precise meaning of them, i.e. that they show the temperature and density at the point with maximum value of the corresponding observable.

We are grateful to the referee for highlighting this issue. We have since added further clarifying text that defines the shapes used within the caption of Fig. 4

---

## Editorial Decision

resubmitted